# The Role of LFA-1 for the Differentiation and Function of Regulatory T Cells—Lessons Learned from Different Transgenic Mouse Models

**DOI:** 10.3390/ijms24076331

**Published:** 2023-03-28

**Authors:** Tanja Klaus, Alicia Wilson, Michael Fichter, Matthias Bros, Tobias Bopp, Stephan Grabbe

**Affiliations:** 1Department of Dermatology, University Medical Center Mainz, 55131 Mainz, Germany; 2Institute for Immunology, University Medical Center Mainz, 55131 Mainz, Germany; 3Max Planck Institute for Polymer Research, Ackermannweg 10, 55128 Mainz, Germany

**Keywords:** LFA-1, Treg, transgenic mouse models

## Abstract

Regulatory T cells (Treg) are essential for the maintenance of peripheral tolerance. Treg dysfunction results in diverse inflammatory and autoimmune diseases with life-threatening consequences. β_2_-integrins (CD11a-d/CD18) play important roles in the migration of leukocytes into inflamed tissues and cell signaling. Of all β_2_-integrins, T cells, including Treg, only express CD11a/CD18, termed lymphocyte function-associated antigen 1 (LFA-1), on their surface. In humans, loss-of-function mutations in the common subunit CD18 result in leukocyte adhesion deficiency type-1 (LAD-1). Clinical symptoms vary depending on the extent of residual β_2_-integrin function, and patients may experience leukocytosis and recurrent infections. Some patients can develop autoimmune diseases, but the immune processes underlying the paradoxical situation of immune deficiency and autoimmunity have been scarcely investigated. To understand this complex phenotype, different transgenic mouse strains with a constitutive knockout of β_2_-integrins have been established. However, since a constitutive knockout affects all leukocytes and may limit the validity of studies focusing on their cell type-specific role, we established a Treg-specific CD18-floxed mouse strain. This mini-review aims to delineate the role of LFA-1 for the induction, maintenance, and regulatory function of Treg in vitro and in vivo as deduced from observations using the various β_2_-integrin-deficient mouse models.

## 1. Treg Subsets and Functions within the Immune System

Regulatory T cells (Treg) are a specialized subgroup of CD4^+^ T cells that can be divided into two main subsets based on their origin. Naturally occurring thymus-derived Treg (tTreg) as well as peripherally induced Treg (pTreg) [1,2]. tTreg have a T cell receptor (TCR) repertoire with high affinity for autoantigens and therefore maintain peripheral tolerance by suppressing autoreactive CD4^+^ T cells [1]. Tregs are distinguished by the expression of the high-affinity IL-2 receptor cell surface chain (CD25) and the transcription factor forkhead box p3 (FOXP3). tTreg development requires activation of the TCR and upregulation of markers like glucocorticoid-induced tumor necrosis factor (TNF) receptor-related (GITR), the TNF family members OX40 and TNFR2, and in some cases CD25. Next, IL-2 and signal transducer and activator of transcription 5 are responsible for differentiation into mature FOXP3-expressing tTreg [3]. In many cases, FOXP3 is a crucial control gene in the development and function of Treg [4,5,6,7,8]. Indeed, humans with a mutation in the *FOXP3* gene lack functionally active Treg and consequently develop immunodysregulation, polyendocrinopathy, and enteropathy X-linked (IPEX) syndrome that is linked to several autoimmune diseases like atopic dermatitis, inflammatory bowel disease, and type 1 diabetes mellitus [5]. In scurfy mice, a loss-of-function mutation of *FOXP3* results in an IPEX-like autoimmune phenotype [9]. 

Unlike tTreg, pTreg develops extrathymically from naive CD4^+^FOXP3^−^ T cells in the presence of transforming growth factor ß (TGF-ß). They are mostly generated by non-self antigen recognition of allergens [2,10]. Unlike tTreg, pTreg lacks the ikaros family member HELIOS as well as neuropilin-1 (NRP1) [11]. In addition, ex vivo induced Treg (iTreg) can be generated in an in vitro culture from naive CD4^+^FOXP3^−^ T cells by the addition of exogenous TGF-ß [12,13]. All these Treg subsets expose their suppressive function in a FOXP3-dependent manner. 

In addition to FOXP3-dependent Treg populations, FOXP3-independent Treg subsets develop in the periphery. These Treg subsets consist of type 1 pTreg (Tr1) and iTr35 [12,14]. Tr1 Treg produce high levels of the anti-inflammatory cytokine IL-10 and co-express the surface markers lymphocyte-activation gene 3 (LAG3) and CD49b [15,16]. iTr35 Treg are induced by IL-35, in contrast to Tr1 Treg, which mediate suppression via IL-35. This unique Treg subset is known to regulate infectious disease [17], as well as transplantation tolerance, and play a crucial role in suppressing autoimmunity [18,19,20]. 

In non-lymphoid tissue, Treg are defined as tissue-resident Treg. These Treg are identified by the expression of the interleukin 33 (IL-33) receptor (ST2) and killer cell lectin-like receptor subfamily G1 (Klrg1), named “tisTregST2”. These ST2-positive tissues Treg express tissue-regenerative factors such as Amphiregulin (Areg) alongside T helper 2 (Th2)-associated factors, including high levels of transcription factor Gata3 [21]. Tissue homeostasis and specific functions of ST2-positive Treg cells have been characterized in different non-lymphoid tissues like visceral adipose tissue (VAT), muscle, colon, and skin [22,23,24,25]. All these Treg subtypes, including both FOXP3-dependent and -independent developed Treg populations, are thought to be essential for maintaining tissue homeostasis and immune tolerance [12,23]. 

Autoimmune reactions are often characterized by a reduced number of functionally active Treg [4]. For example, reduced levels of Treg have been described in psoriatic arthritis [26], systemic lupus [27,28], or Kawasaki disease [4,29]. In vitro studies have shown that Treg confer suppression of effector T cells by a variety of different mechanisms, including anti-inflammatory cytokines (e.g., IL-10, TGF-β) [30,31], induction of apoptosis [32,33], metabolic pathways [34,35], and modulation of the maturation and function of antigen presenting cells (APC) [30,36,37,38,39]. In addition, Treg can support Th responses by reflecting the functions of different Th cell subsets. For example, Th1-like Treg produce IFN-γ, Th17-like Treg generate IL17, whereas Th2-like Treg produce IL-4 [40]. 

In this context, the β_2_-integrin receptor LFA-1 (leukocyte function-associated antigen 1) has been demonstrated to be critical for Treg differentiation and homeostasis [41]. Different transgenic mouse strains with a constitutive knockout of β_2_-integrin receptors have been established and used to characterize the role of β_2_-integrins for T cell and Treg functions. However, because a β_2_-integrin knockout affects all leukocytes and thus limits the ability to delineate cell type-specific roles, we recently established a CD18-floxed mouse strain. However, since a constitutive knockout of -integrins affects all leukocytes, and thereby may limit the possibility to delineate cell type-specific roles, we recently established a CD18 floxed mouse strain. This enabled us to generate mice with a conditional knockout of β_2_-integrins specifically in Treg. This review aims to summarize our knowledge on the role of LFA-1 for Treg induction, maintenance, and function as studied in various β_2_-integrin-deficient mouse models.

## 2. Role of LFA-1 on T Cells 

T-effector cells play an indispensable role in the defense against infections, whereas Treg serve to maintain immune homeostasis. Therefore, targeted migration (or ‘homing’) of T cells into peripheral tissue is important for immune surveillance. β_2_-integrins are surface heterodimeric receptors composed of a variable α chain (CD11a-d) and a constant β chain (CD18) and are expressed specifically by leukocytes [42]. LFA-1 (CD11a/CD18) is the only β_2_-integrin family member expressed on T cells [43]. The α- and β-subunits are each composed of three different domains, including long extracellular domains, transmembrane domains, and cytoplasmic tails [44]. LFA-1 binds intercellular adhesion molecules (ICAMs) expressed by vascular endothelial cells and other leukocytes, especially by APC within the immunological synapse (IS) [7]. LFA-1 also facilitates T cell entry into lymphatic tissues to enable their migration to sites of inflammation and infection [45,46]. Activation-dependent conformational changes determine the binding affinity of LFA-1 [42,47]. At its low-affinity state (bent-closed formation), LFA-1 does not bind to any ligand, thereby enabling T cell circulation through the bloodstream. Chemokines such as chemokine ligand (CCL19) and CCL20, secreted by activated endothelial cells, activate LFA-1 and allow its high-affinity (extended-open) binding to ICAM-1. The integrin conformation change can lead to a 10,000-fold affinity increase of LFA-1 to ICAM-1 [44,48]. This interaction leads to downstream signaling and cytoskeletal rearrangements, which are important for T cell motility [49]. The process through which intracellular signals like chemokines or T cell receptor triggering induce integrin activation and conformational changes in the cytoplasmic tail that favor its extension is called ‘inside-out’ signaling. Similarly, intracellular responses due to integrin adhesion to ligands induce ‘outside-in’ signaling. This leads to the acquisition of a high-affinity state of LFA-1 [47,49]. 

Long-lasting T cell/APC interactions mediated by the high-affinity adhesiveness of LFA-1 are essential for the full activation of T cells [7]. Chen et al. demonstrated strong LFA-1-dependent adhesion between Treg and APC, leading to cytoskeletal rearrangements in the latter and a lethargic state of dendritic cells (DC), which resulted in impaired T cell priming [50], as shown in Figure 1A. These findings supported a LFA-1 contact-dependent suppressive mechanism for Treg [50]. Furthermore, blockade of LFA-1 on polyclonally stimulated iTreg in vitro was shown to induce the differentiation of FOXP3^+^ Treg, while plate-bound anti-LFA-1 antibodies resulted in a reduction of FOXP3 expression [51]. 

Moreover, LFA-1/ICAM interaction influenced human T cell polarization by modulating Th1 and Th2 responses [49,52]. An imbalance between Th1 and Th2 has been associated with several autoimmune diseases, including, for example, multiple sclerosis, rheumatoid arthritis, and type 1 diabetes [53]. In this regard, LFA-1 signaling together with TCR stimulation triggered glycogen synthase kinase (GSK)3β-dependent Notch1 activation by γ-secretase proteolytic cleavage and upregulated t-bet expression, favoring a shift to Th1 cytokine production, specifically of IL-2 and interferon (INF)-γ [53], as depicted in Figure 1B. Notch is known to be a critical differentiation factor for T cell effector function and supports the so-called ‘second touch hypothesis’ [54]. This theory suggested that signaling induced by LFA-1/ICAM interaction fine-tunes the immune response via Notch1 to enhance T cell effector priming and migration to sites of inflammation [49]. Altogether, these findings outline the complexity of LFA-1 signaling to modulate immune responses.

## 3. Leukocyte Adhesion Deficiency Type-1 (LAD-1)

The crucial role of β_2_-integrins for the immune system is emphasized by the phenotype of patients suffering from leukocyte adhesion deficiency type-1 (LAD-1). LAD-1 is a rare disorder that affects only one person per million people worldwide [55]. In LAD-1 patients, a mutation in the CD18 gene leads to attenuated expression of functional heterodimeric β_2_-integrins, resulting in impairment of leukocyte migration into tissue [56]. CD18 expression in some patients differs between cell types and is subdivided into a mild (>30%), moderate (2–30%), and severe form (<2%) [57]. About 75% of patients with severe LAD-1 die by the age of 2 years. Patients with moderate LAD-1 survive childhood, but their mortality amounts to 50% by the age of 40 [58]. Cell type-dependent differential CD18 expression results in a very heterogeneous manifestation of the clinical phenotype, predominantly related to defects in neutrophil migration and pathogen killing functions [59]. 

The most prevalent symptoms are recurrent bacterial infections of the skin, periodontitis, and poor wound healing [60,61,62]. In addition to these immunodeficiency symptoms, LAD-1 patients may suffer from autoimmune phenomena like inflammatory bowel diseases (IBD) and type 1 diabetes or nephritis [63,64]. Thus, clinical symptoms reflect immunodeficiency as well as autoimmunity. Until now, the only curative therapy has been allogeneic hematopoietic stem cell transplantation [65,66].

Patients with wildtype β_2_-integrin suffering from autoimmune diseases like IBD presented with diminished Treg frequencies as described above [4]. It has been demonstrated that LAD-1 patients suffering from intestinal bowel disease (IBD) presented with a higher number of Foxp3^+^ Treg, but these displayed reduced suppressive activity [67]. This finding suggests that β_2_-integrins critically regulate Treg biology. 

## 4. β_2_-integrin-Deficient Mouse Models

The export of functional integrins from the endoplasmic reticulum is dependent on the formation of both α- and β-subunits of the β_2_-integrin heterodimer. In the absence of the β-subunit expression, the α-subunit remains unstable and is degraded [68]. 

Different transgenic mouse models have been established by deleting either the α- or the common β-subunit to understand the role of the given β_2_-integrin(s) for leukocyte functions. In the following chapter we compare the characteristics of the β_2_-integrin-deficient mouse models, focusing on the clinical phenotype and the consequences of LFA-1 deletion on Treg (Figure 2).

### 4.1. CD18 Hypomorphic Mouse (CD18^hypo^)

CD18 hypomorphic (CD18^hypo^) mice were originally established in 1993 [69]. An insertion mutation attenuated CD18 expression by 84–98%. This low-level CD18 expression is comparable to a moderate form of LAD-1 [69,70]. CD18^hypo^ mice display increased neutrophil counts and a reduced immune response to chemically induced peritonitis [70]. When backcrossing CD18^hypo^ mice (129Sv/C57BL/6J background) with PL/J mice, the offspring developed a chronic skin disorder characterized by hyperplasia of the epidermis, subcorneal microabscesses, and T cell infiltrates in the dermis, something very reminiscent of human psoriatic dermatitis [71]. As no infectious organisms were found on lesional skin, and the phenotype could be suppressed by corticosteroids, this skin disorder might be autoinflammation- or autoimmune-driven [70]. Only mice with a homozygous CD18^hypo^ PL/J mutation developed this phenotype, suggesting that the disorder is recessively inherited. However, backcrossing between the susceptible PL/J CD18^hypo^ strain and the resistant C57BL/6 CD18^hypo^ strain resulted in an F1 generation with no phenotype. When backcrossing mice of this F1 generation with CD18^hypo^ PL/J mice, the F2 mice developed skin lesions, suggesting that other modifier genes were responsible for the development of the skin phenotype [71].

The skin disorder in these mice is mainly T cell-driven, as the abundance of activated CD4^+^ T cells secreting high levels of the Th1-type cytokine IFNγ in the skin was increased. Depletion of these CD4^+^ T cells resulted in a complete resolution of psoriatic dermatitis [71]. Furthermore, the increased activation state of CD4^+^ CD25^+^T cells suggested an altered T cell biology. Regarding Treg, CD18^hypo^ PL/J mice had a reduced output of tTreg, and under iTreg promoting conditions, they were rather converted into Th17 cells in vitro and in vivo [72]. Interestingly, cocultures of Treg derived from Balb/c mice and allogenic DC from CD18^hypo^ PL/J mice showed no increase in Th17-like Treg during coculture, indicating that reduced LFA-1 expression on Treg drives the proinflammatory milieu [72]. 

Moreover, reduced CD18 expression impaired cell-cell contacts between Treg and DC, and CD18^hypo^ Treg failed to suppress DC-mediated activation of naïve T cells [73]. Reduced tTreg and iTreg numbers in CD18^hypo^ mice, as well as impaired suppressive activity of Treg converting into Th17-like cells, may contribute to proinflammatory CD4^+^ T cells in lesional skin, thereby driving psoriasiform dermatitis [72].

In line with this, LAD-1 patients have been reported to suffer from psoriatic skin lesions, but skin infections in LAD-1 patients are mainly polymicrobial-driven [74]. As reduced CD18 expression levels have also been identified on leukocytes in psoriasis patients [72,75,76,77], CD18^hypo^ mice constitute an appropriate tool to investigate psoriatic skin disease in the context of attenuated CD18 expression as well as the role of CD18 in chronic inflammatory processes. 

### 4.2. CD18 Null Mouse (CD18^−/−^)

The targeting construct that introduced a hypomorphic allele in the case of CD18^hypo^ mice was also used to insert a replacement mutation to generate the CD18^−/−^ mouse strain with an originally mixed 129/Sv and C57BL/6J background [78]. This construct contains a neomycin-resistance cassette disrupting the 5′ boundary of exon 3, therefore preventing the synthesis of CD18 protein. Thus, CD18^−/−^ mice completely lack β_2_-integrins and share essential characteristics with severe LAD-1 patients, including spontaneous mucocutaneous infections [78]. CD18^−/−^ mice are characterized by chronic dermatitis with facial and submandibular erosions, splenomegaly, and neutrophilic lymphadenopathy [78].

The absence of CD18 in these mice was accompanied by decreased tTreg and pTreg numbers; these Treg populations were devoid of in vitro suppression activity and consequently failed to protect from colitis in vivo [43]. Furthermore, as described above, LFA-1 on T cells is critical for the formation of an immune synapse with APC [7] and is required for the strong adhesion of Treg to DC [50]. Chen et al. demonstrated an indispensable role of LFA-1 on Treg to facilitate adhesion to DC and to prevent conventional T cells (Tconv) binding to DC due to an accumulation of cytoskeletal components within the DC/Treg contact area as an indirect suppressive mechanism [50]. Lack of suppressive function in CD18^−/−^ Treg was also observed in LAD-1 patients, correlating with autoimmune bowel disease in these patients [67].

Interestingly, backcrossing CD18^−/−^ mice to the PL/J strain did not result in psoriasiform dermatitis as described for CD18^hypo^ mice [71]. As mentioned above, pathogenic CD4^+^ T cells might be responsible for the phenotype of CD18^hypo^ mice [71]. CD18^−/−^ CD4^+^ T cells were not able to extravasate to sites of inflammation and therefore could not exert inflammatory effector function in the skin [71,79]. Grabbe et al. confirmed this in a T cell-mediated allergic contact dermatitis model [80]. Injection of hapten-induced effector T cells isolated from hapten-sensitized CD18^−/−^ mice directly into the ears of CD18^−/−^ mice restored the ear swelling phenotype of these mice [80,81]. As a result, while LFA-1 was not required for naïve T cell priming, it was important for their extravasation, as previously reported [80]. 

Hence, β_2_-integrin deficiency manifests a dysregulation of immune homeostasis and contributes to the inflammatory phenotype of CD18^−/−^ mice. These aberrant T cell-mediated immune responses in CD18 null mice suggest that the pathology in LAD-1 patients may not only be due to defects in neutrophils but that T cell defects may contribute to many clinical symptoms [80]. Thus, CD18^−/−^ mice are an appropriate model to study the relevance of β_2_-integrins for the differentiation and function of leukocytes and constitute a reliable model for severe human LAD-1 in the context of chronic inflammation [78]. 

### 4.3. LFA-1 Null Mouse (CD11a^−/−^)

In LFA-1 null mice (C57BL/6J background), both alleles of β_2_-integrin α chain encoding CD11a are disrupted [82]. The gene targeting construct replaces exons 2–6, which encode the signal peptide and the extracellular region of CD11a. Initial characterizations of LFA-1-deficient mice revealed no obvious skin phenotype but splenomegaly and decreased lymph node (LN) size. Lymphocyte frequencies and CD8^+^ T cell cytotoxic responses were normal [82]. Further studies revealed lower Treg numbers in lymphatic organs such as the spleen and LN. In contrast to CD18^−/−^ mice, which had a lower tTreg output [43], LFA^−/−^ mice had higher Treg frequencies/numbers in the thymus compared to wildtype mice [41]. However, LFA-1 appears to be important for Treg induction in the periphery, as LFA-1-deficient mice showed a reduced ability to convert CD4^+^CD25^−^ T cells into functional Treg in peripheral tolerance induction assays. Similar to CD18^−/−^ Treg [43], CD11a^−/−^ Treg failed to suppress T cell activation in vitro and was not able to suppress inflammation in experimental colitis [41]. Therefore, both strains share essential characteristics regarding LFA-1-dependent Treg functions. 

Furthermore, it is proposed that Treg might suppress Tconv activation in two distinct steps: Treg form aggregates with DC in an LFA-1-dependent manner, resulting in active downregulation of CD80/CD86 expression on DC in both a LFA-1- and CTLA-1-dependent way [83]. This suppression of DC maturation prevented antigen-reactive T cells from being activated, as reported in in vitro studies [83]. In agreement, LFA-1 deficiency abrogated the aggregation of Treg with DC. As a result, the absence of LFA-1 on Treg resulted in a decrease in CD80/CD86 expression on DC. The Treg-dependent DC suppression in this context prevented T cells from getting activated [83]. 

Further in vivo experiments with LFA1^−/−^ mice have shown an aggravated course of MOG-dependent experimental autoimmune encephalomyelitis (EAE) as a rodent model of multiple sclerosis [84]. Reduced numbers of Treg in the inflamed CNS of LFA^−/−^ mice correlated with enhanced expansion of autoreactive T cells [84]. In contrast to previous findings [41], a decrease in Treg was associated with a decrease in Treg thymic output [84], emphasizing the importance of LFA-1 in Treg generation and homeostasis. 

### 4.4. CD18Foxp3 Mouse

In the mouse models described so far, all leukocytes lack β_2_-integrins (CD18^−/−^, CD18^hypo^) or LFA-1 (CD11a^−/−^). Thus, it is difficult to assess the cell type-specific role of β_2_-integrins in vivo in the context of LAD-1 pathology. Therefore, we considered it necessary to establish and characterize mice with cell type-specific β_2_-integrin deficiency [42,85,86]. In the course of these studies, we recently generated mice with a FOXP3-specific deletion of CD18. These mice were generated by crossing mice with floxed CD18 (CD18^fl/fl^) [85] with Foxp3^cre^-expressing mice, resulting in a Treg-specific knockdown of LFA-1 (CD18^Foxp3^) as recently described by us [86]. These mice spontaneously develop a dermatitis-like phenotype and systemic, multi-organ inflammation. These findings confirmed the crucial role of LFA-1 on Treg activity. In contrast to CD11a^−/−^ and CD18^−/−^ mice, LFA-1-deficient Treg in CD18^Foxp3^ mice were still able to home into tissues [86]. Interestingly, several tissues presented with higher Treg frequencies than observed in the control animals. These observations suggest that LFA-1 may not be essential for Treg transendothelial migration, as previously described [87]. However, it is also possible that Treg after thymic emigration still displays sufficient amounts of LFA-1 to colonize tissue. Both the time courses of FOXP3 promoter-driven Cre-dependent CD18 deletion in these cells and LFA-1 protein turnover may play a role in this regard. 

Nevertheless, Treg dysfunction, which is highlighted by their attenuated interaction with DC as a prerequisite to inhibit DC activation and to prevent the binding of (naïve) T cells to DC [50,83], might be the main driver of the inflammation in these mice. In vitro studies have shown that LFA-1-deficient Treg form fewer and shorter cell-cell contacts with DC, which may result in increased DC and Tconv activation as observed in vivo. These observations support the data from LFA-1^−/−^ mice, suggesting that LFA-1 on Treg was required for Tconv suppression. 

An x-linked mutation in the *foxp3* gene, as apparent in scurfy mice, caused Treg dysfunction, which resulted in autoimmune multi-organ inflammation as well as the production of anti-nuclear antibodies (ANA) [88,89]. Likewise, the autoimmune phenotype of CD18^Foxp3^ mice was underlined by the presence of ANA in serum that was associated with systemic inflammatory autoimmune diseases such as lupus [90,91], scleroderma [92], and autoimmune blistering skin diseases like pemphigus [93] and bullous pemphigoid [94]. Thus, CD18^Foxp3^ mice share essential characteristics with scurfy mice regarding immune dysregulation and phenotype. 

Altogether, the Treg-restricted, ß_2_-integrin-dependent breakdown in immune tolerance leads to spontaneous multi-organ inflammation in these mice. In contrast to the mouse models described above, CD18^Foxp3^ mice constitute the first murine model to study the functional role of LFA-1 on Treg without the interference of other CD18-deficient immune cells.

## 5. Conclusions

The marked susceptibility of LAD-1 patients to infections has been linked primarily to defects in leukocytosis, particularly of neutrophils. However, much less is known about the higher incidence of autoimmune diseases in LAD-1. In addition to the established role of neutrophils, the role of dysfunctional Treg is sparingly investigated in this context. The mouse models presented here allow researchers to investigate inflammatory and autoimmune processes caused by CD18 deficiency. 

Treg plays an important role in the maintenance of tolerance towards self (tTreg) and environmental (pTreg) antigens. So far, the importance of LFA-1 for T cell activation and polarization has been well established [95]. Moreover, it was demonstrated that loss of LFA-1 led to attenuated Treg induction and Treg dysfunction and thereby dysregulation of immune tolerance, which is mainly characterized by chronic skin inflammation. Thus, CD18-deficient mouse strains share crucial pathological features with FOXP3-deficient scurfy mice. Deficiency of LFA-1 on naive T cells and Treg thereby has a similar negative effect on self-tolerance as the loss of FOXP3, which constitutes a crucial transcriptional key regulator of Treg. Nonetheless, in the case of constitutive β_2_-integrin deficiency, it is difficult to distinguish cell type-specific defects from intrinsic and crosstalk-dependent defects in vivo. Therefore, it is of major interest to generate mouse models with cell-specific deletions of distinct β_2_-integrins. In addition to the mouse line with a Treg-specific LFA-1 knockdown, we have also generated mouse lines with a conditional knockout of β_2_-integrins specifically in CD11c^+^ DC [85] and Ly6G^+^ neutrophils [96]. Both strains enabled the delineation of the cell type-specific role of β_2_-integrins in disease models. Mice with a DC-specific knockdown β_2_-integrins presented with a delayed onset of experimental autoimmune encephalomyelitis as a rodent model of multiple sclerosis and an attenuated course of disease [85]. In a model of pulmonary aspergillosis, mice with a neutrophil-specific knockdown of β_2_-integrins showed an impaired early innate immune response due to various impaired pathogen-killing functions of neutrophils [96]. It will be interesting to generate mouse strains with a conditional knockdown of β_2_-integrins in other leukocyte populations to elucidate their contribution to the complex pathology observed in LAD-1 patients.

## Figures and Tables

**Figure 1 ijms-24-06331-f001:**
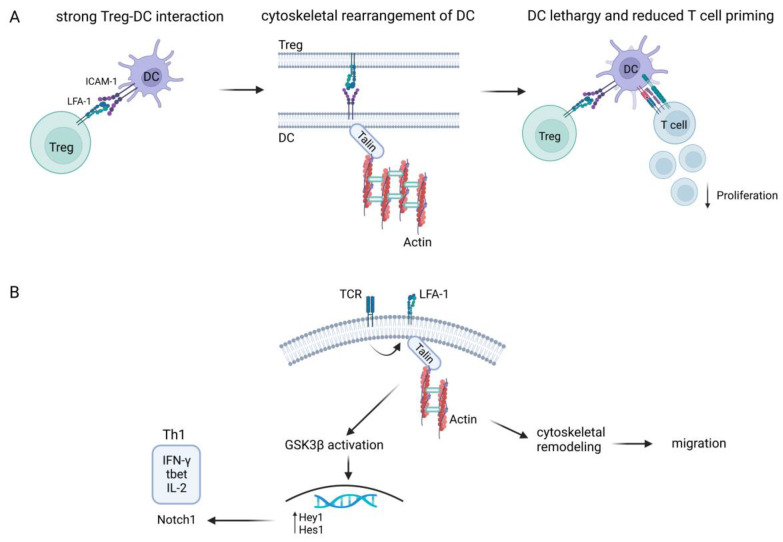
Role of LFA-1-mediated contact for T cell response and suppression. (**A**) LFA-1-mediated suppression of T cells. Long-lasting high-affinity contact between Treg and DC via LFA-1-ICAM-1 interaction activates the cytoskeletal rearrangement of Fascin-1-dependent actin polarization in DC. This conformational change causes DC lethargy, preventing T cell proliferation and activation. This mechanistic control of T cell activation emphasizes Treg-mediated DC suppression via LFA-1 contact [50]. (**B**) Th1 response in T cells is driven by LFA-1 signaling. TCR inside-out signaling leads to the binding of Talin-1 to the β_2_-tail of LFA-1, enabling its high-affinity activation. Following this, LFA-1 triggers downstream signaling pathways that control cytoskeletal rearrangement and the migration of activated T cells. LFA-1 also induces glycogen synthase kinase (GSK)3β-dependent Notch1 activation and the production of Th-1-related cytokines and transcription factors [7,52,53] (created with BioRender.com (accessed on 7 March 2023).

**Figure 2 ijms-24-06331-f002:**
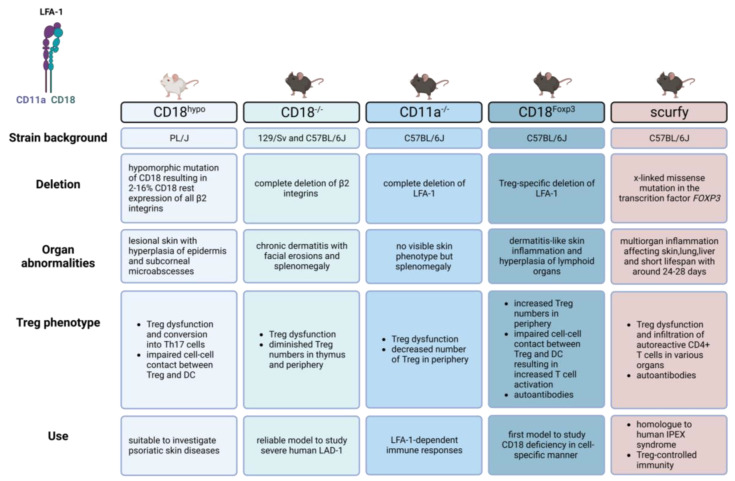
Integrin-deficient mouse models in comparison to foxp3-deficient scurfy mice (created with BioRender.com (accessed on 7 March 2023).

## Data Availability

Not applicable.

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
