# Peer review of "The Role of LFA-1 for the Differentiation and Function of Regulatory T Cells—Lessons Learned from Different Transgenic Mouse Models"

_ijms, 2023, doi:10.3390/ijms24076331_

Round 1
Reviewer 1 Report
Manuscript titled "Understanding the role of LFA-1 on regulatory T cells -lessons..." is a very nice work. Apart from some minor additions that I think should be there, I don't see any more bugs.
Below are my comments.
1. Maybe I would slightly change the title.
2. The end of the abstract is to be rewritten.
3. β2-integrin deficiency - is it needed in keywords?
4. The end of the last paragraph should be slightly improved and the aim of the work much more emphasized.
5. The whole work is quite well described - very nice graphics.
Summary is to be rewritten. They're hard to read. Please correct.
Author Response
We thank the reviewers for the very helpful comments on our manuscript.
We have addressed all issues raised by the reviewers. For clarity, we have written our responses to the reviewers in blue and the corresponding alterations of the manuscript in red (attached word dokument).

Reviewer 2 Report
Here are few comments I think you can make a part of your review:
1. Authors could also mentioned more recent markers/methods for cheracterization of Tregs.
2.To tie the worb back into public health need, it may be wise for the authors to use a brief sentence or two to describe LAD incidence:
a. only affects 1 in 1e6 people
b. Mortality for severe leukocyte adhesion deficiency-1 was reoprted as 75%by the age of 2 years.
c. Patients with moderate diseases (2% to 30% CD18 expressing neutrophls) survive childhood, with multiple infections affecting the skin and mucosal surfaces: documented mortality exceeds 50% by the age of 40 years.
3. Re: conclusion, could potentially talk about how cell specific deleations can be generated (using LysCre mice for myeloid-specific knockouts; LckCre for T cells, etc.
Author Response

(The authors gave the same response as above.)
